# The Battles around Urban Governance and Active Citizenship: The Case of the Movement for the Caracol da Penha Garden

**Jorge Gonçalves**

CiTUA—Center for Innovation in Territory, Urbanism and Architecture, Instituto Superior Técnico, University of Lisbon, 1049-001 Lisbon, Portugal; jorgemgoncalves@tecnico.ulisboa.pt

**Abstract:** Building a civil society that can act as a collaborative voice in the processes of change that take place in the territories does not always come about naturally and peacefully but is often something that needs to be won, based on concrete episodes in daily life. These changes are framed by the ongoing trend of shift from a form of power carried out in accordance with the old values of opacity, autonomy and imposition to one dominated by transparency, informality and sharing. The consolidation of governance processes in line with the legitimate exercise of local, national, regional or metropolitan government therefore makes sense. A descriptive methodology is adopted here of the process of affirmation of an organic movement of citizens, identifying step by step the interactions between actors that led to the reversal of the initial decision taken by the municipality. This emblematic case is framed theoretically by the ongoing paradigm shift related to the modes of exercising power at the local scale. This analysis of the case of the Movement for the Caracol da Penha Garden in Lisbon, Portugal is a contribution to understanding how this slow and difficult transformation takes place in urban and metropolitan environments and how the learning that can be taken from these processes can be of great benefit to all urban stakeholders.

**Keywords:** active citizenship; urban governance; green public spaces; public participation; urban power; urban policy

## 1. Introduction

Cities and metropolitan areas are home to an enormous and growing complexity in terms of institutions, competencies and jurisdictions, which make their management inefficient, bureaucratic and tendentially closed to the equally complex universe of civil society, which is understood as a diffuse constellation of individuals and formal and organic organisations, which voluntarily act in favour of specific sectorial or territorial interests [1].

Since these two universes—public administration and civil society—have very distinct dynamics and characteristics, when they collide a potential for change can be identified that should be analysed with a view to understanding the significance and depth of the adjustment that urban and metropolitan planning and management mechanisms can potentially undergo, as indeed argued by [2].

More so than in the planning phase, where the outlines of projects are not yet very clear, it is during the phase of announcement of the type of intervention, its location and its characteristics that sometimes generates certain agitation that calls into question not only the design project itself and the methodology applied but also the entire political and technical power structure that gave rise to it. Gardens, cycle paths, public space, new buildings, mobility, equipment, services and infrastructures are the most common reasons that trigger such collision processes between universes [3–5].

This article aims to be a useful contribution to understanding the slow but troubled and difficult shift from a classical hierarchical power structure—that of government—to the emergence of new and complementary structures of shared power—governance—as argued by [6].

As this change is much more complex and demanding than it seems, it is believed that it is worthwhile to describe the case of the Movement for the Caracol da Penha Garden, while framing it in a context of the emergence of a new power culture, as [7] refer to it, and, within that context, the protagonism of a civil society that does not resort only to protest, but reveals in some cases that it is more sophisticated, agile and cunning [8] than the bureaucratic powers that be, which also raises risks and apprehension [9].

The methodology adopted is mainly descriptive, based on a reading of the process of protest against public powers in the light of the changes taking place in the power structures that lead to greater protagonism for organic social movements. The elements of analysis were obtained from the information gathered directly from official sources and indirectly through documentation published by independent and credible media, as well from the interviews conducted with the citizens' movement.

The article is organised into four sections in addition to this introduction, focusing on the signs that reveal the changes in the exercise of power in Section 2 and the role of civil society in that change in Section 3. Section 4 describes the process brought about by the Movement for the Caracol da Penha Garden and the section that follows it aims to compare the results with the theoretical framework proposed in the initial sections; it also presents the main conclusions.

## 2. The Changing Universe of the Exercise of Power

The whole environment of change that is being felt in politics, and also in society, the general circumstances and even the economy, has had implications for the way territories are organised and managed. Ref. [10] puts forward two reasons why there is a crisis in politics:

- The first point he makes about this crisis is that policy is not playing its role well, with a failure to reform existing policy with a view to make it more effective;
- The consequences of a failure to adapt to new issues related with shared assets are more complicated, where the administration does not have an institutional level of decision-making that is adequate or legitimate.

It is here, according to [10], that the shift from government as a traditional way of dealing with problems, very hierarchical, very closed, etc., to a style of governance, the concept for which serves to "refer to new forms of governance within and beyond the nation-state", is born (p. 6).

Accordingly, one agrees with and follows Innerarity's ideas [11] when he argues that "a reticular world requires relational governance. Networks require more complex instruments, such as trust, reputation, and reciprocity. These new constellations require institutional innovation in governance processes and go beyond classical administrative routines. The new governance suggests a form of coordination between political and social agents characterised by regulation, cooperation, and horizontality. In complex societies, models and procedures for governing cannot aspire to a form of unity that cancels out diversity; governing means managing heterogeneity" (pp. 10–11).

Green spaces should directly be a part of this discussion because they have been the source of many of the controversies in the urban context and have given rise to many of initiatives and actions from civil society ([12], either for social reasons (green spaces as a place for leisure, socialising, etc.), for environmental reasons (green spaces as a strategic place to combat the effects of climate and changes thereto, which can be particularly grave in urban areas), or for financial reasons (as these spaces are almost always associated with an improvement in the urban image and real estate valuation) [13].

These aspects have led to the formation of several civic movements documented by the media that have progressively given visibility to the new dynamics. The creation of gardens and redevelopment of others, proposing innovative green spaces and linking such spaces in a network, among other things, have served to ensure that this topic has so far had an important influence on the aforementioned context of change, which is registered in

what Timms and Heimans designate as the transition from "old power values" to "new power values" [7].

This shift, which is slow and troubled, as indeed the tension and polemics between those who wish to be heard and considered and those who have the legitimate power to decide show, is being fostered in part by a demand for new and better-quality public spaces, almost always in the form of gardens.

Accordingly, it is believed that the transformation of an old style of power (authoritarian and closed) into one that recognises people's right not only to vote in cyclical elections, but also to participate in the decisions that affect them in their daily lives, is beginning to take place, albeit very slowly. In other words, governance and green spaces have been a central tool for these achievements, which now involve new stakeholders that once appeared almost invisible. Ref. [14] illustrates well the complexity that is possible today when dealing with urban public green spaces, when he put the place-based governance in the middle of the relationship between users, green space managers and urban green spaces

However, as Molin also reminds us, the inclusion of new actors in matters related to the public space and, as in the case at hand, to green spaces, must not only mean asking and demanding for more and better, but should also integrate them and their responsibility into the decision-making process, the solution and even, perhaps, the maintenance and management thereof [14]. This is what one can truly call a collaborative process—joint decisions and joint responsibility. However, this process of the complexification and widening of the range of stakeholders involved in urban interventions is now widespread, as is well illustrated by [15].

### 3. Civil Society's New Weapon: Active Citizenship

*3.1. Roots and Features*

The renewed central role of civil society in urban and metropolitan governance processes was not left out of the New Leipzig Charter, the transformative power of cities for the common good ([16], which explicitly references inclusive and sustainable development goals for cities and metropolitan areas, or even the Sustainable Development Goals [17], which so often emphasise sustainable and inclusive development [18,19].

Accordingly, achieving a rethink with regard to the coexistence of a public universe and a universe marked by a dynamic civil society may provide an opportunity to confront old problems and even new challenges, such as those emerging now with urban renewal, climate change, the ageing population, or the invasive nature of technology.

In modern societies, the concept of citizenship originates from the understanding that individuals are members of a community, and in a democracy are apt, in legal and practical terms, to participate in the exercise of political power through electoral processes, whereby all individuals are equal before the law and have equal rights and duties [20].

Ref [21] argues that citizenship rights should not be seen so much in evolutionary terms, but rather as a pattern of concentric circles in which new rights appear over a certain core of already established fundamental rights. The inner circles contain the civil rights, political rights, and social rights, and in the outer circles one can see the emerging rights, the new rights that have been attributed to citizens since the 1970s.

It is possible to define these new rights as a set of measures on which the reform of the Public Administration has been based and which can be grouped into four major aspects:

- Administrative simplification, aimed at optimising relations between the administration and citizens;
- Improved qualification and motivation of the agents involved in the whole process;
- Changes in the power structure and organisation, for example, through decentralisation and delegation of powers;
- Establishing mechanisms for the participation of citizens in Public Administration, providing greater approximation between the two, and an adaptation of administrative responses to certain problems. The individual thus acquires the ability to influence the diagnosis and decision-making process, as well as the way in which administrative

measures are applied. Furthermore, democracy and the social equity of administrative measures and regulation of the citizen's daily life are consolidated [20].

The changes in this relationship can be seen as an expansion of citizens' rights in a modern democracy. What is being dealt with is a more active citizenship with a more diversified range of participation rights, which may translate to the gaining of new citizenship rights [22].

This discussion focuses on the idea of a more just city, which, beyond the live and passive matters that serve as resources for urban development, requires governance that actively includes citizens in driving their own destinies and management thereof. A stage can be reached when policies can promote social sustainability and meet the challenges of inclusion challenges that are general or common to most cities, and even as far as believing that successful communities are those that can reinvent local citizenship [23].

Ref. [24] is of the opinion that the traditional concept of bureaucratic organisation and hierarchical coordination is defined by characterising areas of action, relationships between superiors and subordinates, process control and centralisation. Included in this model is the idea of power and authority, the extent of which varies and is distributed along the hierarchical chain. This set of ideas, common in modern societies, has been called into question by the increasing difficulties in terms of affirmation:

- reduced flexibility in the decision-making process;
- the absence of incentivisation to control costs;
- the lack of transparency;
- the almost total lack of accountability and innovation associated with the development of a culture that is more concerned with procedures than with performance.

But what does public participation have to offer in the current situation? According to [25], it offers a means of:

- Incorporating citizens' values, interests, perspectives, aspirations and needs into decisions that affect them;
- Contributing to a broader knowledge of both problems and opportunities and possible options/alternatives in the urban space;
- Improving decision-making—sustainable decisions require consensus and the integration of different perspectives on the problem; it is not enough that the solution is technically feasible, it must also be environmentally, economically and socially viable.

### 3.2. Performative Dimensions of Citizenship

There is a tendency to expand the body of stakeholders involved in decision-making processes concerning the urban space, and in particular public space, as a central and growing concern of citizens. There is still some way to go between the acceptance of these changes and their implementation that is difficult to overcome today. The difficulty has to do with the inability of the formal and legitimately constituted powers to deal with the voices that want to be heard [26]. Fung also highlights the risk, given the interests that need to be expressed, of the collective interest being manipulated and even skewed if the state gives up its role as arbitrator, decision-maker and guarantor of democracy.

Given these cultural, but also practical issues, a look at collaborative processes that deal with recognition of the role of citizen participation in building sustainable urban futures is warranted. The vision of the sustainable city, presented namely in the New Athens Charter, can only be implemented "through the joint efforts of all stakeholders in sustainable urban management and planning processes" [27], requires a new approach that is not merely process governance, i.e., based on formal state institutions, but urban governance, i.e., based on the involvement of citizens and their civic movements as a means of ensuring social cohesion. Starting out with a pyramidal hierarchy, the horizontality of relationships and contacts is now valued in an acceptance of co-accountability and co-decision.

The need for the involvement of citizens and active citizenship to shape this new understanding of urban governance concepts has been encouraged by supranational organisations such as the OECD, UN, World Bank, to name just a few [28].

Accordingly, there is a need to migrate to a more enabling and stimulating governance, making the most of the transforming potential of active citizenship. In the case of public spaces, and specifically green spaces, their physical diversity, the cultural diversity of citizens, citizens' ways of using and valuing these spaces and the diversity of how they self-organise, are not compatible with generic "one-size-fits-all" policies [29].

Collaborative processes manifest themselves in two different ways:

- The first is in the relationship between citizens and their organisations, which were already in existence or were formed with the intent to address a specific topic. This relationship can be called horizontal, as no hierarchies can be identified among its components. Indeed, this would seem to be a great asset in the success of these movements [30];
- The second is in the relationship between the formal powers (elected, democratic and bureaucratic) and citizens and their representative movements, which can perhaps be called vertical, as the power status of one of the parties almost always manifests itself (even if only through imposition and financing) [31].

This process of change between cultures of power feeds into many of the initiatives emerging from the sphere of active citizenship and aimed at realising and even expanding the right to the city [32]. This corresponds to the exercise of a capacity for use, intervention and transformation assumed by the local communities affected, for example, by unjust or poorly justified decisions or, as [33] puts it, "The right to the city is not merely a right of access to what already exists, but a right to change it after our heart's desire" (p. 939). It can emerge out of formal (pre-existing) or organic (reactively constituted) movements [34].

These processes are usually moments that inspire new models of governance in the relationship with other entities, but also the creation of considerable social capital that reinforces local identities, cohesion and solidarity [35]. In the specific case of citizenship aimed at urban transformation, the complexity of the processes and the difficulties in obtaining concrete results in a timely manner are the greatest challenges that these movements face.

Urban forms of active citizenship [36] accordingly include experience of very concrete issues and challenges and clearly defined stakeholders and targets. These are processes of resistance and transformation with a focus on public spaces, housing and mobility, among other topics, as the central issues in the promotion of active citizenship and of a new power relationship.

The following stage is how the processes are carried out, or, as [37] puts it, the performative dimensions of citizenship or acts of citizenship [38].

As will be seen below, the transition can be difficult, in terms of allowing for the possibility of delegation of powers from one party to another, with the local authority retaining its oversight responsibility.

## 4. The Movement for the Caracol da Penha Garden

### 4.1. Background

The Caracol da Penha Garden, in Lisbon, seems to have everything to illustrate what has been said and discussed in the previous sections, such as, for example, participatory citizenship or governance of public green spaces, among other aspects. It is also a very recent case and not yet over, so there are certain restrictions in relating the history.

Reconstituting the process was made more difficult by the lack of legal/formal documents both from the side of the Lisbon City Council—it was not possible to access the minutes of the executive meetings—and the Lisbon Municipal Mobility and Car Parking Company (EMEL). Credible alternative sources were found in the Lisbon Municipal Assembly, on the website of the Movement for Caracol da Penha Garden and diverse various media and social networks. Figure 1 shows the interest levels that the issue has originated

since June 2016, when the case of the challenge to EMEL's proposal for a new car park was reported.

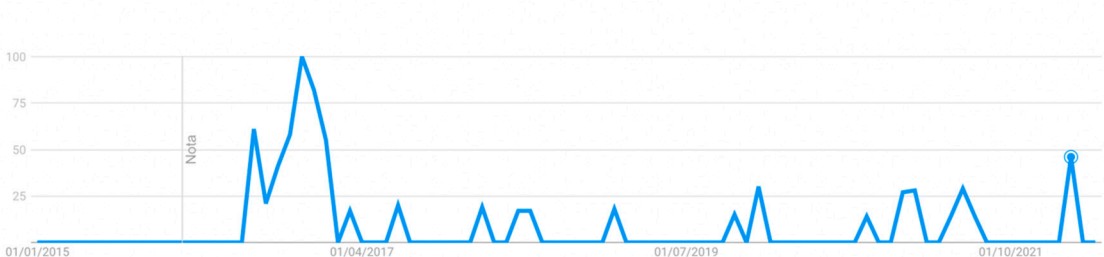

**Figure 1.** Web search for the term "Caracol da Penha" from January 2015 to May 2022. Source: Google Trends (accessed on 30 April 2022).

At any rate, the case of contestation of a car park by residents who were not previously organised and where many did not even know each other, and who, on top of that, wanted to replace it with a garden, is sufficiently interesting and pertinent to try to devalue all the obstacles encountered.

The local citizens' participation, the reversal of the decision of CML and EMEL, the use of the *Orçamento Participativo* (Participatory Budget) mechanisms, the ability to produce news, the organisation of the information and the ways in which it was communicated, among many other aspects, make this an emblematic case and an example for many others, as well as a learning opportunity for all stakeholders involved.

### 4.2. The Site

The place where it all took place is in Lisbon, close to Avenida Almirante Reis, one of the city's structural roads that leads directly into the city centre, where there was once an important market for supplying the people of Lisbon with what they needed. This avenue served as a channel of entry for products into Lisbon for the said market. It is accordingly marked by elements considered much more rural than the other structural road in the city centre that is Avenida da Liberdade.

The property in question, owned by the Lisbon City Council, is within a block of houses, measures almost 10,000 sq. m in surface area and is somewhat difficult to access from Rua Marques da Silva (Figure 2).

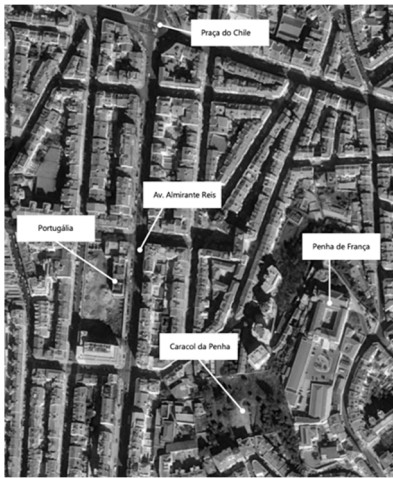

**Figure 2.** Location of the site and relationship with surrounding landmarks.

### 4.3. The Imposition of a Decision

Using Google Earth's Chronology tool, it was possible to reconstruct the history of the Caracol da Penha from 2001 to the end of 2019 and verify that it has remained practically the same throughout the two decades of the 21st century. The only exceptions are the demolition of some precarious structures located to the east between 2007 and 2014 and of some buildings of a sounder appearance; the image dates from 2016, giving the idea that the "clearing" was carried out very close to that year (Figure 3).

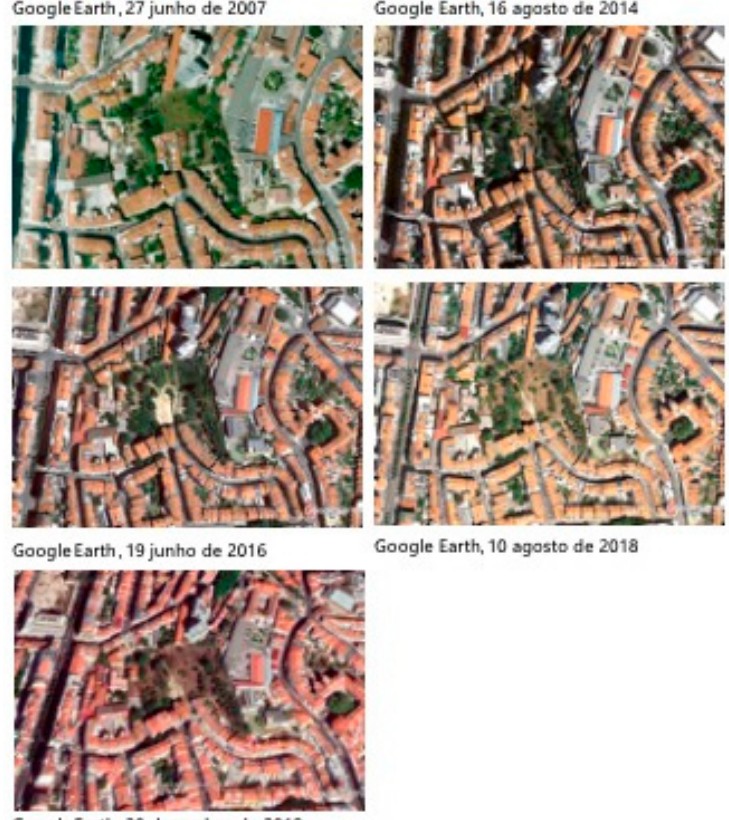

**Figure 3.** The Caracol da Penha site between 2007 and 2019. Source: Google Earth (accessed on 15 April 2021).

This slight modification that occurred in 2016 was already a sign that things were soon to change for this historically abandoned piece of land.

Information on parking in Lisbon also goes some way to explain the changes on this land: "In 2017, Lisbon City Council (CML) intends to have 30,000 more paid parking spaces on public roads throughout the city than there were at the end of last year. In all, almost 82,000 spaces should be marked off by then, in line with the target defined in the management contract between the Council and Lisbon Municipal Company for Parking and Mobility (EMEL) ( . . . ). ( . . . ) In January, when the change to the regulations for the area was opened up to public discussion, Alvalade, Arroios and Penha de França were some of the places indicated both by Manuel Salgado, alderman for Urbanism and the new proposal's signatory, and EMEL" (Diário de Notícias, 13 April 2016).

Penha de França was already referred to in April 2016 as a privileged area where EMEL would expand its activities at the request of residents. This reasoning for the profound change in the car parking management policy was also described in the press article: "EMEL and CML have had frequent requests from the resident population and parish councils to intervene in areas that are not covered by the current regulations", explained the municipal company, stressing that "residents are the one who suffer most from the

parking disorder in those areas where there is no EMEL concession" (Diário de Notícias, 13 April 2016).

The clearing of the land that could be seen on 19 June 2016 can now be better understood with the news that the aim was to expand the supply of regulated and charged parking in some civil parishes, namely Penha de França. Once again, the Diário de Notícias newspaper is used to better understand what contribution to the volume of regulated and paid parking in Lisbon the Caracol da Penha site would make: "The project currently under way provides for three 'platforms' on the hillside, which has a fairly steep slope. The upper platform will feature a crèche, a children's playground, a kiosk and a lookout point with a café terrace. The two lower platforms will be used for parking, which will total 86 spaces (99 were initially planned). Also, according to the company, the new parking spaces should be available by the end of the summer" (Diário de Notícias, 27 June 2016).

The very same proposal, with a few differences (for example, the crèche has now disappeared), features in the 21 October 2016 report drawn up by the Municipal Assembly, which defended the initial project in response to the citizens' "We need a garden" petition, which had already announced the challenge to the decision of the public authorities. While acknowledging the healthy exercise of active citizenship, the Lisbon Municipal Assembly recommended in the said report the rapid start of the works to "meet the parking needs of residents and visitors". At this stage, the project had not yet been opposed by the two parish councils affected—Penha de França and Arroios.

The strategy followed by CML and EMEL and announced in early 2016 provided the backdrop to the decision taken in June 2016 with the aim of creating, with the support of the parish councils involved (and, as it turned out, the Lisbon Municipal Assembly), 86 more parking spaces on an abandoned plot of land in Penha de França, which ended up triggering a local protest and the whole subsequent process.

The not-so-open discussion of how everything happened has led me to include this process under what has been referred to as 'old power values', i.e., with the provision of limited information, not backed up by concrete data (on the lack of parking and residents' demand for more parking) and little openness to real discussion with the residents. It is the typical top-down imposition, where only government and no governance can be observed. This was, thus, the beginning of the Movement for the Caracol da Penha Garden.

*4.4. The Contestation*

An act of imposing something on someone is often met with a reaction [39]. Here too, this phenomenon seems to have taken place, albeit first in the form of contestation. The fact that less than one fifth of the whole project area was dedicated to people, who lacked open and free spaces in a highly densified and urbanistically consolidated area, was enough to see the emergence of the protest that was based on the following notion described in the website of the Movement for Caracol da Penha Garden: "The CML/EMEL project's mistake: to reserve only 17.81% of the existing space for use by the people!" In the civil parishes of Penha de França, where there is only 0.8 sq. m of space per inhabitant, and Arroios, where that ratio is 1.2 sq. m per inhabitant (the two are the worst-off civil parishes in Lisbon in this respect), such a mistake could be fatal.

But while this was the fundamental reason for contesting the project for the creation of 86 parking spaces, the arguments were also based on the guidelines laid out in the Municipal Master Plan (PDM) in force in Lisbon for the site—namely in the Risk Chart, which makes reference to the problem of increasing impermeable surface areas both with respect to an increase of surface water flows and soil erosion and compaction, as the groundwater can no longer be renewed, leading to rupture of the soil, opening holes in the streets and leading to landslides. Accordingly, the land, with its steep slope, was classified as being of "Moderate risk, with an elevated risk of movement of masses".

Finally, a look at the Land Classification Chart in the PDM showed that the site was classified as "green space for recreation and growing". Indeed, the residents claim that there is an intense photographic record, which goes back to the early 20th century, that

shows that the land was intensively and methodically cultivated. Today, in other forms, it remains a space where things are grown: besides the various existing fruit trees (including plum trees, vines, peach trees, lemon trees, avocado trees, olive trees and banana trees), there are still some small vegetable gardens maintained by residents from the surrounding neighbourhoods. In the future, as was enshrined in the PDM, protection of this wealth should be the goal, as well as a guarantee that the following aim be fulfilled: to maintain the land as a true "green space for recreation and growing".

Whilst these were the tools to contest the 86 parking spaces and the little attention given to people, it was never clear how the protest actually started, and when and by whom. One thing is clear: it was started by residents supported by an excellent communication strategy and the dissemination of their initiatives, but also, as seen above, supported by documents of good technical quality, in which they related their convictions.

On its website, the Movement justified its existence as follows:

"The Movement for the Caracol da Penha Garden emerged out of the desire of residents of the civil parishes of Arroios and Penha de França to see the creation of a real public garden in a green space that they already knew well from looking at it from their windows ( . . . ). The Movement, as soon as it was born, began to grow. And continues to grow! It brings neighbours and friends together, friends of neighbours and neighbours of friends! Now, people who did not know each other, but met daily in the streets, come together to achieve a common goal for the community: they want a real public garden in an existing green space! They want the regeneration of this space and its opening to the public, for collective enjoyment.

The Movement for the Caracol da Penha Garden is made up of people of all ages, all education levels, all professions, and areas of study. We are men, we are women, we are people of all genders, we are children, teenagers and old people and we reflect total diversity" (in https://www.caracoldapenha.info/quem-somos) (accessed on 23 May 2022).

*4.5. The Reaction (Or How to Come Up with Alternatives)*

4.5.1. Proposal 180 of the Participatory Budget

Following the protest, i.e., the announcement that the residents did not accept the project that was being imposed on them, it was necessary to come up with an alternative proposal that would be viable and serve as a basis for negotiation with CML.

The "We need a garden" petition sent to the Lisbon Municipal Assembly on 13 September 2016 with more than 2600 signatures already included not only the criticism of the CML/EMEL original plans, but also provided the basis for the proposal of what the future of that space should be in the view of the residents. Basically, their aim to replace a landscaped car park with a real garden. Here are the seven reasons they gave:

1. Because the green space already exists. It just needs to be redeveloped and opened to the public.
2. Because there is no other real garden in this area. This is one of the most densely populated areas of Lisbon and one of the urban areas with the least green space, with no public garden distanced from car circulation routes.
3. Because this is the last chance. There are no other large green spaces in this part of the city that are still free.
4. Because the space already belongs to everyone. The land in question already belongs to the Lisbon City Council, so the process of creating a public garden is simplified and costs are lower.
5. And nature is fundamental. Contact with nature is fundamental for health and well-being. This could be a space where children can run, play, and have fun in safety, where adolescents can do sport at their leisure, a space where older people can enjoy good times and socialise close to home.

6.  Because imagination is the limit. This area of the city has few public spaces that enable people to socialise in the open air. This garden will allow residents—and others—to think up new things and let their imagination run wild: community vegetable gardens, tai-chi, sports championships, swings, multipurpose areas, café terraces, picnics, trees, shade and birds, small concerts . . .

7.  And a garden prevents flooding. Flooding is a recurrent problem in Lisbon. A garden mitigates this (in https://www.caracoldapenha.info) (accessed on 23 May 2022).

These arguments already announced a strategy of proposing to the 2016 Participatory Budget the creation of a new garden, which was realised in the form of Proposal 180-Caracol da Penha Garden.

The "Caracol da Penha Garden" proposal was also accompanied by an intense campaign to mobilise the population for the voting process that followed. One of the most emblematic examples occurred on 27 October 2016 when residents and artists came together for a cultural programme for the Caracol da Penha Garden (Cultural Programme for a Garden), which included about 20 performances during the afternoon and evening of that day. It featured street theatre, yoga, music, and dance in a day with programming for all ages (Figure 4).

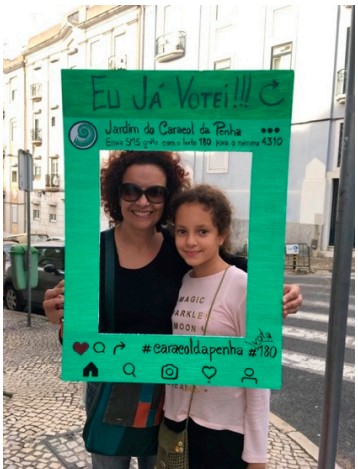

**Figure 4.** The pride of having voted for the garden in Lisbon's Participatory Budget. Source: https://www.caracoldapenha.info/ (accessed on 30 December 2021).

The positive result came about, with the additional feat of it being the project that received the most votes ever in the Lisbon Participatory Budget, with a total of 9477 votes.

The success of the Participatory Budget's Proposal 180-Caracol da Penha Garden, made it possible to reverse Lisbon City Council's decision and to raise EUR 500,000, with a forecast of its implementation within 24 months.

4.5.2. The New Project for the Caracol da Penha Garden

The next step was to define the landscape project that would meet the expectations of all those who voted and mobilised for the garden. To this end, the Movement for the Caracol da Penha Garden launched a participatory process to define the future space and named that process "How do you imagine the Caracol da Penha Garden?", whereby all people would send in suggestions in the form of images, documents or links that could clarify what they wanted.

The idea was that after gathering this information, the design team would propose an initial version of the garden, which would then be publicly discussed again. The participatory process in the definition of the Garden Project generally followed this path (Figure 5):

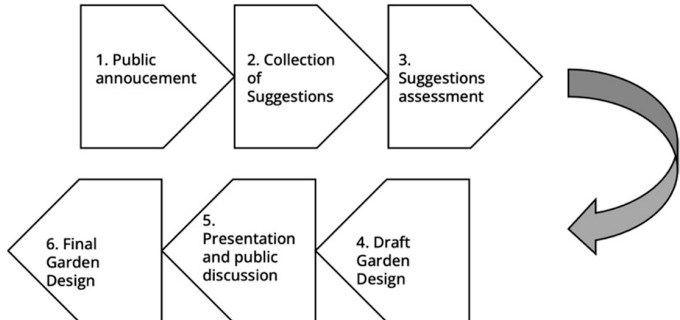

**Figure 5.** Participatory process stages in the definition of the Garden.

The governance process was once again based on the involvement of citizens in the process of co-defining the functional programme for the garden, thus ensuring coherence in the Movement's actions since its spontaneous formation. From these efforts, and the collaboration of landscape architects, the new design for the Caracol da Penha Garden took shape. Lisbon City Council was also an important stakeholder in the process, as it had to not only finance the project and the works, but also validate the solution determined in the context of the collective interest and more global interests of the city of Lisbon.

In September 2017, the hope was that the garden could be open in early 2019, something that did not take place. Formalities and approvals at the City Council level and months of work with architects and designers meant that it was only in June 2019 that the final design was completed.

On 9 May 2019, the contract for the construction of the Garden was signed and the works started in November 2019, which then experienced considerable delays due to the restrictions imposed by the pandemic. The garden is now expected to be opened in 2022.

## 5. Results and Conclusions

This section does not aim to add much to the reflection made at the beginning of this text and the description of the emblematic case of the movement for the Caracol Garden of Penha. It is, however, worthwhile to highlight the mechanisms by which the change of values in the exercise of power took place.

In this specific case, the decision to allocate a particular use to public property was taken in a conventional way, i.e., top-down, based on arguments that were found to be not very robust, as requests from residents for more parking spaces were never shown.

This classic form of decision-making was challenged by residents who questioned the use assigned to the space and suggested an alternative use. At first, the local government rejected this suggestion, and it took an organised, creative, dynamic and constructive civic movement to reverse the initial decision.

The systematised description of the actions of the Movement for the Caracol da Penha Garden, when articulated with the discussion initially made about the ongoing processes related with a change in the power culture, going from the idea of opaque government, which excludes and does not share, to a form of governance that emerges as an aggregator, and the emergence of a civil society that is not only made up of formal and long-lasting bodies, but also of organic, spontaneous and short-lived movements, has given rise to a set of important ideas that I would like to emphasise.

First of all, there is the difficulty of the classic hierarchical powers to review their decisions reached without adequate argumentation and this naturally hinders capacity for discussion with the stakeholders involved; then, there is leadership from civil society that is capable of activating a robust protest and resistance to a decision that seemed definitive and irreversible, while staying within the rules of democracy; then comes the capacity of this new organic movement to react and build a viable alternative to a car park, going against an institutional decision, using the participatory budget mechanism to its advantage and mobilising local stakeholders for their goals and in new ways; finally, there is the reversal of the public entities' initial decision, and even the inclusion of the

Movement in the subsequent design and programming process for the Garden, as well as in the request made by Lisbon City Council to operate and manage the café in the new green space.

All this shows how the processes of political change do not occur naturally, but have to be won, using a lot of energy, availability and resources of all kinds; it is also true that these processes constitute useful and non-reversible learning moments towards the deepening of active citizenship.

What this section also seeks to demonstrate is that the shift in paradigm in the exercise of power identified by Heimans and Timms (2018) [7] is already quite visible in the urban and metropolitan context, particularly in cases associated with public spaces or even related with transformations derived from redevelopment and regeneration processes.

But this contribution also clarifies that there is a process underway towards a more collaborative and, at the same time, more responsible and engaged society, as [7] describe.

The enormous energy required to reverse a decision already taken, driven by a community base that is normally excluded from decision-making processes, shows that the classic power system only accepts the revision of its positions under great pressure. However, each time it does so, it brings us closer to the level of citizen power as defined in Arnstein's Ladder of Participation [40].

Far from revealing the political crisis that Innerarity talks about [10], these spontaneous, temporary, and focused movements can be fundamental in confronting and combating the negative externalities generated in the democratic system on the local scale, in the form of abstention and populism, for example. But whilst one of the catalysts of this Movement has been technology—the use of digital platforms, social media, etc.—technology can also be used in negative ways, so it is important to continue to study how smart governance can be a key process for the survival of collaborative democracy in the future.

This is an emblematic case of the tension between civil society and its political representatives in the paradigm shift of urban and metropolitan management. It is believed that framing and describing the process of the Caracol da Penha Garden has aided in understanding how imperative it becomes:

- More transparency in the argumentation and decision taken by local governments;
- A more genuine encouragement of community involvement in the stages of problem identification and the identification of possible solutions;
- A renewed capacity of local government to convene relevant actors for co-decision and even co-responsibility for the implementation and management of the chosen solution.

Each of the above points is a complex world, but it is urgent to put them into practice in order to give more legitimacy and sustainability to urban intervention processes. In other words, what should and can be drawn from this process is that building a truly active citizenship, whereby a participatory democracy is an imperative, does not have a clear path ahead that is free of difficulties. However, it is considered that these difficulties are natural in such a profound transition between very different ways of looking at the exercise of power in an urban context, namely with the valorisation of governance processes where the actors and agents who best help identify problems, design solutions and, in some cases, follow-up or assume co-responsibility in their implementation are involved.

Success in this process of change will also mean a giant step towards the sustainability of urban and metropolitan spaces in their different faces, namely in the adoption of more collaborative planning and urbanism, fundamental for the construction of more socially resilient communities.

**Funding:** This research received no external funding.

**Institutional Review Board Statement:** Not applicable.

**Informed Consent Statement:** Not applicable.

**Conflicts of Interest:** The author declares no conflict of interest.

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
