# Peer review of "The Battles around Urban Governance and Active Citizenship: The Case of the Movement for the Caracol da Penha Garden"

_sustainability, doi:10.3390/su141710915_

Round 1

Reviewer 1 Report

This is a very decent manuscript that tells an interesting descriptive success story about public participation processes in urban governance in specific settings. This can become a paper of substantial interest for the readers of Sustainability.

I suggest thpublication with some very minor revisions.

1) Although the English of the text is overall quite good I do have issues with the use of the term "migration" in this manuscript, see abstract and lines 234 onwards. I did not really find similar examples in the literature. The term migration implies a physical movement or displacement of living organism and as such is not really applicable to power shifts that are more abstract in nature. Perhaps the term "shift" would be more appropriate* Or some other term?

2) These are the concluding words of the manuscript (lines 543 onwards): "In other words, what should and can be drawn from this process is that building a truly active citizenship, whereby a participatory democracy is an imperative, does not have a clear path ahead that is free of difficulties" I would very much want to author to expand on this a little further.

Author Response

Sustainability | manuscript 1857424 | Reviewer 1

Before detailing each of the comments and giving a suitable response to them, the author would like to thank the reviewer for their dedication and for the high quality and acuteness of their comments. This quality helped to transform and profoundly improve the original text. I hope you agree the new version of the manuscript 1857424 – Sustainability.

Reviewer 1

Author’s responses

This is a very decent manuscript that tells an interesting descriptive success story about public participation processes in urban governance in specific settings. This can become a paper of substantial interest for the readers of Sustainability.

I suggest the publication with some very minor revisions.

Many thanks for your positive assessment

1) Although the English of the text is overall quite good I do have issues with the use of the term "migration" in this manuscript, see abstract and lines 234 onwards. I did not really find similar examples in the literature. The term migration implies a physical movement or displacement of living organism and as such is not really applicable to power shifts that are more abstract in nature. Perhaps the term "shift" would be more appropriate* Or some other term?

The reviewer has totally reason. We have corrected in abstract “migration” by “shift” and in line 234 by “change”

2) These are the concluding words of the manuscript (lines 543 onwards): "In other words, what should and can be drawn from this process is that building a truly active citizenship, whereby a participatory democracy is an imperative, does not have a clear path ahead that is free of difficulties" I would very much want to author to expand on this a little further.

We agree with the reviewer and because of that we add these two paragraphs: “However, it is considered that these difficulties are natural in such a profound transition between very different ways of looking at the exercise of power in an urban context, namely with the valorisation of governance processes where the actors and agents who best help identify problems, design solutions and, in some cases, follow-up or assume co-responsibility in their implementation are involved.

Success in this process of change will also mean a giant step towards the sustainability of urban and metropolitan spaces in their different faces.”

Reviewer 2 Report

Dear Authors,

I would like to thank the author/s for their effort.

Several efforts have been made by the authors to examine and analyze the case of the movement for the Caracol da Penha Garden in Lisbon, Portugal, in order to provide an insight into the way this kind of movement plays out in urban and suburban environments. But there are some minor comments that need from authors to modify it to enhance the research:

1- As the abstract is so brief, it does not demonstrate the methodology (which is descriptive as mention in Introduction) used in this study. To give the reader a sense of what to expect at the end, I prefer the authors to include a brief summary of the findings.

2- In general, it is preferable if the authors can avoid usage of first-person pronouns such as “he” “We,” for example in line (78) page (2), etc… as much as possible to maintain the tertiary nature of this publication and maintain a neutral voice in the article.

3- In page (6), author forget to provide the page number of what Harvey (2004) said in line 238-240.

4- There are so many blurs in figures 4 and 5 and they need to be resized so they are easier to read.  

5- In page (9) line (317), there is a mention of those (...). (...) that do not understand.

6- In my opinion, the reader will not be able to understand the differences that mention by the author in page (10) line (340) when said ‘ The very same design’ without add maps for previous and current version to discover the differentiation.

7- It would be helpful if the author focused more on the appropriate process stages that can lead to a right decision at the end by government and civil society, to provide a role model for how that can happen.

By the end, I want to say that you did a great job and all this comments is to encourage the author to enhance the quality of paper.

Thank you for your efforts.

Best Wishes

Author Response

Sustainability |  manuscript 1857424 | Reviewer 2

Before detailing each of the comments and giving a suitable response to them, the author would like to thank the reviewer for their dedication and for the high quality and acuteness of their comments. This quality helped to transform and profoundly improve the original text. I hope you agree the new version of the manuscript 1857424 – Sustainability.

Reviewer 2

Author’s responses

I would like to thank the author/s for their effort.

Several efforts have been made by the authors to examine and analyze the case of the movement for the Caracol da Penha Garden in Lisbon, Portugal, in order to provide an insight into the way this kind of movement plays out in urban and suburban environments. But there are some minor comments that need from authors to modify it to enhance the research

(…) By the end, I want to say that you did a great job and all this comments is to encourage the author to enhance the quality of paper.

Many thanks for your positive assessment

1- As the abstract is so brief, it does not demonstrate the methodology (which is descriptive as mention in Introduction) used in this study. To give the reader a sense of what to expect at the end, I prefer the authors to include a brief summary of the findings.

We agree with the reviewer. We add the following text in the abstract: “A descriptive methodology is adopted here of the process of affirmation of an organic movement of citizens, identifying step by step the interactions between actors that led to the reversal of the initial decision taken by the municipality. This emblematic case is framed theoretically by the ongoing paradigm shift related to the modes of exercising power at the local scale”

2- In general, it is preferable if the authors can avoid usage of first-person pronouns such as “he” “We,” for example in line (78) page (2), etc… as much as possible to maintain the tertiary nature of this publication and maintain a neutral voice in the article.

We’ve made all the changes proposal by the reviewer.

3- In page (6), author forget to provide the page number of what Harvey (2004) said in line 238-240.

We have now provided the page number.

4- There are so many blurs in figures 4 and 5 and they need to be resized so they are easier to read.  

Done.

5- In page (9) line (317), there is a mention of those (...). (...) that do not understand.

Sorry, but we don’t understand the comment because line 317 is blank.

6- In my opinion, the reader will not be able to understand the differences that mention by the author in page (10) line (340) when said ‘ The very same design’ without add maps for previous and current version to discover the differentiation.

We understand the reviewer opinion. We’ve changed “design” by “proposal”.

7- It would be helpful if the author focused more on the appropriate process stages that can lead to a right decision at the end by government and civil society, to provide a role model for how that can happen.

The content of the article can also be read the other way round, i.e. as a way of realising that authoritarian and uncontradicted decision making may not have sufficient legitimacy.

However, in order to provide a more robust response to the reviewer, the following text has been added in section 5:

“This is an emblematic case of the tension between civil society and its political representatives in the paradigm shift of urban and metropolitan management. It is believed that framing and describing the process of the Caracol da Penha Garden has aided in understanding how imperative it becomes:

- More transparency in the argumentation and decision taken by local governments;

- A more genuine encouragement of community involvement in the stages of problem identification and the identification of possible solutions;

- A renewed capacity of local government to convene relevant actors for co-decision and even co-responsibility for the implementation and management of the chosen solution.

Each of the above points is a complex world, but it is urgent to put them into practice in order to give more legitimacy and sustainability to urban intervention processes.”

Reviewer 3 Report

The article is poorly written, with english language and style requiring extensive editing.

“The methodology adopted is mainly descriptive, based on a reading of the process of protest against public powers in the light of the changes taking place in the power structures that lead to greater protagonist for organic social movements.” (…) “Reconstituting the process was made more difficult by the lack of legal/formal documents both from the side of the Lisbon City Council – it was not possible to access the minutes of the executive meetings - and the Lisbon Municipal Mobility and Car Parking Company, (EMEL).”
As far as I am understanding the author was not an active part of the process nor was able to collect all the information, in particular from the municipal authorities that had the most active and relevant role in all the process.

“This section does not aim to add much to the reflection made at the beginning of this text and the description of the emblematic case of the movement for the Caracol Garden of Penha. It is, however, worthwhile highlighting the mechanisms by which the change of values in the exercise of power took place.”
So, what was the article for? The results and discussion don’t add anything new to the field and the author admits that.

The article is being published in MDPI - Sustainability but not once sustainability is mentioned in the article. Colaborative planning and ground breaking initiatives like the study case present is important for sustainable governance and the creation of sustainable communities and neighbourhoods in every sense of the word.

Author Response

Sustainability |  manuscript 1857424 | Reviewer 3

Before detailing each of the comments and giving a suitable response to them, the author would like to thank the reviewer for their dedication and for the high quality and acuteness of their comments. This quality helped to transform and profoundly improve the original text. I hope you agree the new version of the manuscript 1857424 – Sustainability.

Reviewer 3

Author’s responses

The article is poorly written, with english language and style requiring extensive editing.

Thanks for the comment. Before being submitted to the journal, this text was completely revised by a native English translator (Liam Burke) who always does translation work for this research group. I would also point out that in this case we have not had any other comments of this kind nor did the reviewer provide any concrete examples that could help us understand what he meant.

“The methodology adopted is mainly descriptive, based on a reading of the process of protest against public powers in the light of the changes taking place in the power structures that lead to greater protagonist for organic social movements.” (…) “Reconstituting the process was made more difficult by the lack of legal/formal documents both from the side of the Lisbon City Council – it was not possible to access the minutes of the executive meetings - and the Lisbon Municipal Mobility and Car Parking Company, (EMEL).”
As far as I am understanding the author was not an active part of the process nor was able to collect all the information, in particular from the municipal authorities that had the most active and relevant role in all the process.

What was mentioned by the reviewer is true, but it should also be underlined that it was considered that the information gathered, directly from official sources and indirectly through documentation published by independent and credible media, as well as the interviews conducted with the citizens' movement, was considered sufficient and adequate in the light of the objectives of the article, and in no way limited its results.

We added in section 1 the following text:

“The elements of analysis were obtained from the information gathered directly from official sources and indirectly through documentation published by independent and credible media, as well from the interviews conducted with the citizens' movement.”

“This section does not aim to add much to the reflection made at the beginning of this text and the description of the emblematic case of the movement for the Caracol Garden of Penha. It is, however, worthwhile highlighting the mechanisms by which the change of values in the exercise of power took place.”
So, what was the article for? The results and discussion don’t add anything new to the field and the author admits that.

We accept the criticism, of course, but we do not agree with it. What we intended to reinforce with that paragraph is that the construction of the ongoing change in the exercise of power on an urban and metropolitan scale can and should be illustrated with concrete actions that are interesting to know and critically evaluate.

In addition, in section 5 some paragraphs have been added to make clearer what we believe must be improved in the relationship between public authorities and civil society, facilitating a paradigm shift that can no longer be halted.

The added text was:

“This is an emblematic case of the tension between civil society and its political representatives in the paradigm shift of urban and metropolitan management. It is believed that framing and describing the process of the Caracol da Penha Garden has aided in understanding how imperative it becomes:

- More transparency in the argumentation and decision taken by local governments;

- A more genuine encouragement of community involvement in the stages of problem identification and the identification of possible solutions;

- A renewed capacity of local government to convene relevant actors for co-decision and even co-responsibility for the implementation and management of the chosen solution.

Each of the above points is a complex world, but it is urgent to put them into practice in order to give more legitimacy and sustainability to urban intervention processes.In other words, what should and can be drawn from this process is that building a truly active citizenship, whereby a participatory democracy is an imperative, does not have a clear path ahead that is free of difficulties. However, it is considered that these difficulties are natural in such a profound transition between very different ways of looking at the exercise of power in an urban context, namely with the valorisation of governance processes where the actors and agents who best help identify problems, design solutions and, in some cases, follow-up or assume co-responsibility in their implementation are involved.

Success in this process of change will also mean a giant step towards the sustainability of urban and metropolitan spaces in their different faces, namely in the adoption of a more collaborative planning and urbanism, fundamental for the construction of more socially resilient communities.”

The article is being published in MDPI - Sustainability but not once sustainability is mentioned in the article. Colaborative planning and ground breaking initiatives like the study case present is important for sustainable governance and the creation of sustainable communities and neighbourhoods in every sense of the word.

We understand the criticism and in the revised article this limitation was, we believe, overcome with explicit references to the concepts pointed out by the reviewer. 

Round 2

Reviewer 3 Report

The article clearly improved and the author completely understood my arguments. The section added at the end of the article, in my opinion, does in fact makes a big difference and rounds up everything that the case study of Penha's Caracol evidences.

I have to admit that I'm not still 100% satisfied with the sustainability part but what was added cover the basics. I'm aware that nowadays the concept of sustainability is reaching a status that everything, everywhere has to mention it, but I do believe that when it comes to colaborative planning, sustainability is extremely important in more ways than one.